# Rapid Agrichemical Inventory via Video Documentation and Large Language Model Identification

**DOI:** 10.3390/ijerph22101527

**Published:** 2025-10-05

**Authors:** Michael Anastario, Cynthia Armendáriz-Arnez, Lillian Shakespeare Largo, Talia Gordon, Elizabeth F. S. Roberts

**Affiliations:** 1Department of Health Sciences, Northern Arizona University, Flagstaff, AZ 86011, USA; lgs82@nau.edu; 2Escuela Nacional de Estudios Superiores Unidad Morelia, Universidad Nacional Autónoma de México, Morelia 58190, Mexico; cynthia_armendariz@enesmorelia.unam.mx; 3Department of Anthropology, University of Michigan, Ann Arbor, MI 48109, USA; trgordon@umich.edu (T.G.); lfsrob@umich.edu (E.F.S.R.)

**Keywords:** agrichemical identification, large language models, avocado production, exposure assessment

## Abstract

Background: This technical note presents a methodological approach to agrichemical inventory documentation. It complements exposure assessments in field settings with time-restricted observational periods. Conducted in Michoacán, Mexico, this method leverages large language model (LLM) capabilities for categorizing agrichemicals from brief video footage. Method: Given time-limited access to a storage shed housing various agrichemicals, a short video was recorded and processed into 31 screenshots. Using OpenAI’s ChatGPT (model: GPT-4o^®^), agrichemicals in each image were identified and categorized as fertilizers, herbicides, insecticides, fungicides, or other substances. Results: Human validation revealed that the LLM accurately identified 75% of agrichemicals, with human verification correcting entries. Conclusions: This rapid identification method builds upon behavioral methods of exposure assessment, facilitating initial data collection in contexts where researcher access to hazardous materials may be time limited and would benefit from the efficiency and cross-validation offered by this method. Further refinement of this LLM-assisted approach could optimize accuracy in the identification of agrichemical products and expand its application to complement exposure assessments in field-based research, particularly as LLM technologies rapidly evolve. Most importantly, this Technical Note illustrates how field researchers can strategically harness LLMs under real-world time constraints, opening new possibilities for rapid observational approaches to exposure assessment.

## 1. Introduction

Agricultural workers in regions with high pesticide and agrichemical use, such as Michoacán, Mexico, often experience elevated exposures to hazardous chemicals, which can impact health outcomes and contribute to environmental degradation [1]. Pesticides have been documented to harm non-target species, contaminate ecosystems, air, water, and soil, as well as cause cancer and damage to multiple organs [2]. The agricultural sector is considered to be one of the most hazardous sectors of work, and toxic agrichemical exposures are a serious concern [3,4,5]. It has been estimated that around 25 million agricultural workers experience an episode of unintentional, acute pesticide poisoning annually [6]. The health of farmworkers in avocado production settings can also be compromised by viruses or occupational exposure to pesticides and other agrichemical products [7]. In agricultural settings, workers may not always perceive these hazards as cause for concern and are often unaware of the full range of chemicals to which they are exposed [8]. In addition, avocado producers may prepare complex agrichemical mixtures while being unable to recount all of the ingredients without the discarded labels/containers present. The disjuncture between which chemicals are present in agricultural work environments and worker/producer awareness can complicate methods of agrichemical exposure assessment that rely on self-report, the validity of which is presumed to be poor outside of licensed applicators and mixers.

In Michoacán, exposure to organophosphates (insecticides), pyrethroids (insecticides), fungicides, and herbicides has been documented among avocado workers [1]. Assessing these exposures is critical, yet challenging, particularly in time-limited settings where rapid data collection is required. Traditional ethnographic methods prioritize prolonged observation and detailed documentation [9], such as in-depth interviews and focus groups [10]. Life history calendars (LHCs) have been used to facilitate retrospective data collection regarding the timing, occurrence, and sequencing of self-reported exposures [11,12,13,14,15,16,17]. The method is thought to enhance retrospective reporting by reflecting the structure of autobiographical reporting [18]. However, these behavioral methods of exposure assessment are reliant upon participant recall, and can be complemented by observation-based inventories of agrichemicals currently in use and/or recently used on a farm. In such cases, primary raw data that may be observed to generate an inventory can take many forms, including handwritten notes, photographs, and video/audio recordings, which can help extract hard to capture information [19].

The method described in this article was part of a larger ethnographic data collection effort concerning situated exposures in avocado farming in Michoacán, Mexico. The method presented here builds on observational methods for assessing exposure without reliance on self-reported data for current year exposures. The method can be used to complement and/or cross-validate more time-consuming methods of exposure assessment involving structured behavioral instruments that rely on self-report, open-ended data collection techniques, and/or biospecimen collection and analysis. By using a short video walkthrough of an agrichemical storage shed, followed by LLM-assisted categorization, the method quickly generates a comprehensive list of visible agrichemicals. The methodological innovation uses an LLM-assisted identification approach to address challenges associated with agrichemical documentation. The LLM-assisted approach is particularly useful in this context, as it has the capability to assist in various aspects of data analysis, such as generating and verifying codes, and illustrating results [20]. This approach enabled the collection of preliminary data without extensive on-site chemical handling, addressing the practical constraints of time, access, and safety.

## 2. Materials and Methods

In May of 2024, the ethnographer (first author) joined a team from the Universidad Nacional Autónoma de México, Campus Morelia to conduct site visits at various avocado farms in Tingambato in Michoacán, Mexico. This included visits to farms that practiced conventional and organic farming methods. The ethnographer took a 55 s video of the agrichemical shed. This included taking a video of bottles and bags containing agrichemicals, where the agrichemical containers on each shelf were rapidly captured Via video. The video was recorded on an Iphone 12 at 1170 × 2532 pixel resolution under indoor mixed artificial and natural lighting conditions, and screenshots were generated from the video at two-second intervals.

The text of the original prompt that was provided by the ethnographer in the field stated: “I am going to upload a series of photos across different chat windows. The photos were taken in an agrichemical storage shed at an avocado farm in Michoacan, Mexico. I need you to help me identify agrichemicals across all the photos that I am going to upload across multiple windows. I want you to sort everything you identify into the following categories: (1) Fertilizers, (2) Herbicides, (3) Insecticides, (4) Fungicides, (5) Other. Do you understand? Do you have any questions about this task?”

Following the visit, snapshots of the video (see Figure 1 for examples) were created at two-second intervals, yielding 31 images. The images were fed in batches of 8 photos per dialog window to ChatGPT, which was employed to identify and categorize visible agrichemicals as fertilizers, herbicides, insecticides, fungicides, or other for each batch of photos. At the time the method was trialed, this was the maximum number of photos that the LLM could handle to produce data for the team in a timely manner. The team then collated the lists to summarize all agrichemicals identified by ChatGPT in each category. The list was produced during the same day that the visit occurred, and the video was captured. The chemical list produced by ChatGPT is illustrated in Table 1.

## 3. Results

The method’s validation, achieved through human verification of ChatGPT’s outputs, demonstrated 75% accuracy in categorizing the chemicals captured in the video (see Table 2). The process involved watching the video of agrichemicals, pausing at relevant points, and taking snapshots of each chemical to cross-check with the AI-generated list. Discrepancies were easily resolved through internet searches, particularly by using visual search tools and referencing Mexican agrichemical company websites. It took approximately 10 h to validate the chemical list generated by ChatGPT for the purposes of this manuscript. This method ensured a more accurate and reliable list. This LLM-assisted technique shows significant promise as an approach that can complement exposure assessments.

Upon review of these findings, ChatGPT recommended several enhancements to improve the reliability and applicability of this LLM-assisted method for agrichemical identification. First, to address “hallucination”—a common issue where LLMs may misidentify or invent chemicals—future iterations of the method could employ prompts that specifically instruct the LLM to “identify only chemicals visibly labeled in the image, without assumptions based on partial or blurred text.” Additionally, refinement of categorization accuracy could be achieved by using prompts that ask the LLM to “first identify the type (fertilizer, herbicide, insecticide, fungicide) based on specific visual and textual cues before proceeding to detailed categorization.” Finally, integrating prompts that invite the LLM to cross-reference detected agrichemicals with common regional products, when possible, may enhance relevance and reduce the likelihood of misclassification. These adjustments provide a basis for further developing this method, allowing future researchers to leverage LLMs with improved precision in similarly restricted fieldwork contexts.

## 4. Discussion

In a 55 s video walkthrough of an agrichemical shed, we subsequently captured screenshots at two-second intervals, yielding 31 images. A LLM (ChatGPT, 4o) was then employed to generate a categorized list of visible agrichemicals under the headings of fertilizers, herbicides, insecticides, fungicides, or others. The method’s validation, achieved through human verification of ChatGPT’s outputs, demonstrated 75% accuracy in categorizing these chemicals. This now dated LLM-assisted technique nonetheless offers promise for data gathering to complement observational exposure assessments in field-based research and environmental monitoring of pesticide use in rural settings. Most importantly, this Technical Note illustrates how field researchers can strategically harness LLMs under real-world time constraints, opening new possibilities for rapid observational approaches to exposure assessment.

This method of rapid agrichemical documentation using LLM assistance has several limitations that impact its reliability and reproducibility. First, the accuracy of chemical identification is constrained by the quality and clarity of the video and images. Low-resolution screenshots can obscure labels and brand names, leading to potential misidentifications or omissions by the LLM, as was observed in the 25% error rate in this study. Second, LLMs like ChatGPT may “hallucinate” results [21,22,23], identifying substances or categories that do not correspond to actual labels. As a Technical Note, this manuscript is intended to demonstrate the feasibility of the approach rather than provide full-scale model validation; future work should build upon this preliminary step with more rigorous testing of different batch sizes and prompt designs tailored to specific LLM versions. Importantly, we continue to apply this approach to rapidly document and categorize complex agrichemical mixtures in agricultural field environments where time constraints of avocado producers make a traditional inventory impractical. Although human validation can correct some of these errors, it requires time and expertise, potentially offsetting the time saving benefits of the LLM-assisted approach that would likely occur in the context of other methods that are more time intensive. Additionally, LLMs may have limited knowledge of specific agrichemical brands or region-specific products, which could lead to misclassification, especially in specialized agricultural settings like those in Michoacán. The approach presented here relies on a brief visual survey rather than a comprehensive chemical inventory, potentially missing chemicals that were not captured within the limited timeframe of the video and/or chemicals that were not contained in the storage shed. In addition, the chemical appearing in a storage shed does not imply it is currently being used but provides strong evidence that the chemical has or will be used at the farm setting where it was documented. Future applications of this method would benefit from more robust image-capturing techniques and from integrating supplementary chemical identification tools to improve accuracy and reduce reliance on subjective verification. These images were taken at one time during the year and given the temporal variability in avocado production cycles, the chemical inventory would likely vary if taken at another time during the year.

### Limitations

This study has several limitations. First, misclassifications or omissions most likely occurred due to blurred or partially obscured text on containers, limited recognition of certain region-specific agrichemical brands, and the use of an earlier model version of ChatGPT that may be less adept than newer iterations. At the time that work for this Technical Note was produced (May 2024), ChatGPT could not adequately process the video files that were directly uploaded, so screenshots were extracted and provided iteratively. The maximum number of images the model could handle in a single chat window in a timeframe amenable to the team’s timeline was eight, which determined our batch size. Future research should systematically test different batch sizes to evaluate whether accuracy improves or diminishes across varying input levels. Finally, this Technical Note reflects a method trialed in the field in 2024 using ChatGPT-4o. As LLM technology evolves rapidly, current and future models (e.g., GPT-5) likely perform differently and more effectively than the version available at the time of data collection. Our approach, however, demonstrates one way that field researchers can interact with and harness LLMs to accomplish tasks that were otherwise difficult under real-world time constraints.

## 5. Conclusions

This rapid identification method builds upon behavioral methods of exposure assessment, facilitating initial data collection in contexts where researcher access to hazardous materials may be time limited and would benefit from the efficiency and cross-validation offered by this method. We emphasize that the goal of this Technical Note is methodological demonstration, and its contribution lies in outlining a rapid, feasible procedure that can inform and complement subsequent, more comprehensive studies. Further refinement of this LLM-assisted approach could optimize accuracy in the identification of agrichemical products and expand its application to complement exposure assessments in field-based research.

## Figures and Tables

**Figure 1 ijerph-22-01527-f001:**
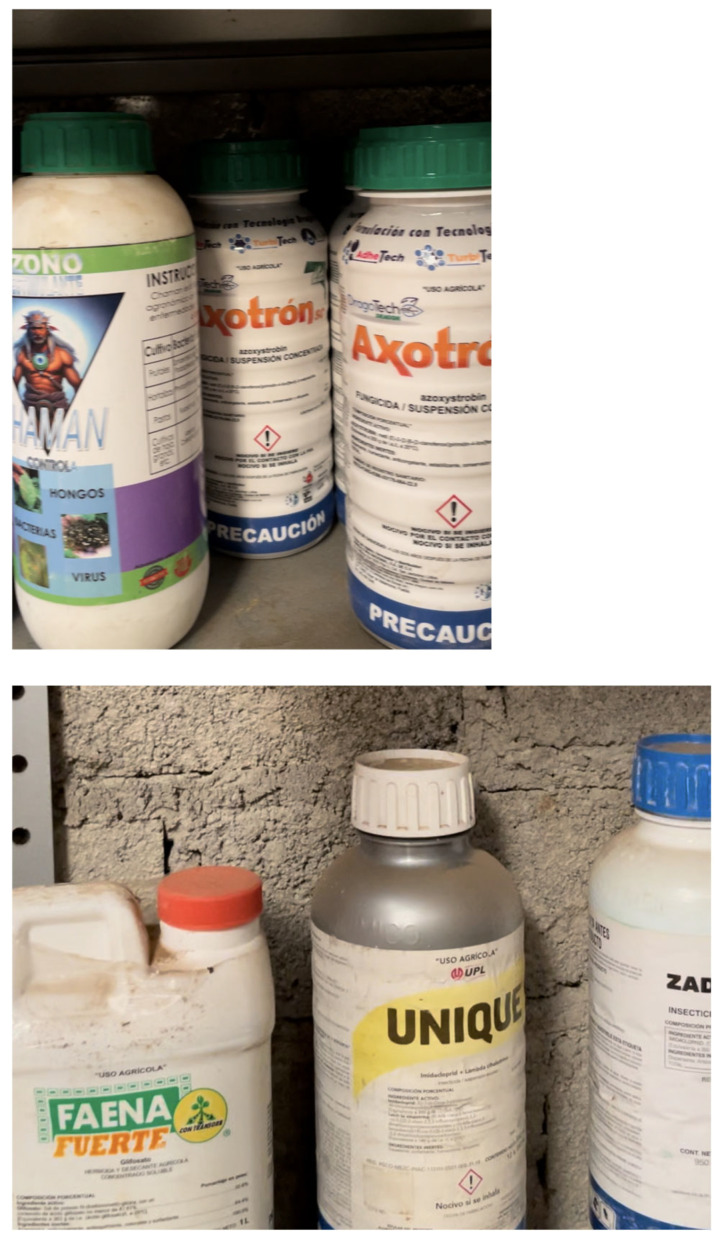
Representative screenshots from agrichemical storage shed video used for LLM-assisted identification.

**Table 1 ijerph-22-01527-t001:** Agrichemicals identified by ChatGPT.

FERTILIZERS Fertilizante Soluble AHUACATL Fermil (various compositions) Fertilizante Soluble MACRO Valgo 50 WG (Contains humectants, dispersants, and other inerts)
HERBICIDES Faena Fuerte (Glyphosate)
INSECTICIDES Fyfanon (Malathion) Vantex (Gamma-Cyhalothrin) Karate (Lambda-Cyhalothrin) Engeo (Thiamethoxam and Lambda-Cyhalothrin) Intrepid (Methoxyfenozide) Cincuate Perkilon (Permethrin) Zelk (Acetamiprid) Sivanto (Flupyradifurone) Karaté Zeon (Lambda-Cyhalothrin) Cintari (Abamectin) Cinchalio (Cyhalothrin) Zadok 25 ST (Thiamethoxam) Unique (Imidacloprid + Lambda-Cyhalothrin) Imida-K (Imidacloprid) ZADOK 350 SC (Imidacloprid) Imida Krone (Imidacloprid) Impiremax (various compositions)
FUNGICIDES Amistar (Azoxystrobin) Revus (Mandipropamid) Propi Ridomil Gold (Metalaxyl-M) Hidromet (Copper Hydroxide) Citlalli 70 WG (Mancozeb and Metalaxyl) Axotron SC (Azoxystrobin) Chaman
OTHER Adigo (Multi-purpose, primarily as a soil treatment) Voga (Contains modified polyether trisiloxane, adjuvant) Valgo (Unspecified ingredients, possibly a wetting agent) BIO HIDROXIDO DE CALCIO (Possibly used for pest control and as a disinfectant) Soluble Calcium (Likely used to correct calcium deficiencies in plants)

**Table 2 ijerph-22-01527-t002:** Human verification of chemicals identified by ChatGPT.

Chemical Identified by ChatGPT	Chemical Identified Through Human Verification (0 = No, 1 = Yes)
FERTILIZERS	
Fertilizante Soluble AHUACATL	1
Fermil (various compositions)	1
Fertilizante Soluble MACRO	1
Valgo 50 WG	1
HERBICIDES	
Faena Fuerte (Glyphosate)	1
INSECTICIDES	
Fyfanon (Malathion)	1
Vantex (Gamma-Cyhalothrin)	1
Karate (Lambda-Cyhalothrin)	1
Engeo (Thiamethoxam and Lambda-Cyhalothrin)	1
Intrepid (Methoxyfenozide)	1
Cincuate	1
Perkilon (Permethrin)	1
Zelk (Acetamiprid)	1
Sivanto (Flupyradifurone)	0
Karaté Zeon (Lambda-Cyhalothrin)	1
Cintari (Abamectin)	1
Cinchalio (Cyhalothrin)	0
Zadok 25 ST (Thiamethoxam)	0
Unique (Imidacloprid + Lambda-Cyhalothrin)	1
Imida-K (Imidacloprid)	1
ZADOK 350 SC (Imidacloprid)	1
Imida Krone (Imidacloprid)	1
Impiremax (various compositions)	1
FUNGICIDES	
Amistar (Azoxystrobin)	0
Revus (Mandipropamid)	0
Propi	0
Ridomil Gold (Metalaxyl-M)	0
Hidromet (Copper Hydroxide)	1
Citlalli 70 WG (Mancozeb and Metalaxyl)	1
Axotron SC (Azoxystrobin)	1
Chaman	1
OTHER	
Adigo	0
Voga	1
Valgo	1
BIO HIDROXIDO DE CALCIO	1
Soluble Calcium	0

## Data Availability

The images and video files analyzed in this study contain potentially sensitive and identifying information about avocado producers and storage practices. For this reason, the underlying data are not publicly available but may be shared upon reasonable request to the corresponding author.

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
