# Peer review of "Rapid Agrichemical Inventory via Video Documentation and Large Language Model Identification"

_ijerph, 2025, doi:10.3390/ijerph22101527_

Round 1

Reviewer 1 Report

Comments and Suggestions for Authors

Review for IJERPH 3716276

Rapid agrichemical inventory via video documentation and 2 large language model identification

This paper is timely. LLMs can provide a quick assessment of chemical inventories through photographs, making it adaptable to a range of conditions, such as this use of a storage shed. However, while the overall approach is simple (nothing wrong with that) and sound, it is not rigorous enough to actually test the model. There are two major concerns: 1) the authors only tried the method by uploading 8 photos, but we don’t know if that was appropriate; the authors should have tried a range of the number of photos (e.g., 1, 2, 4, 8, 15, 31) and reported on how/if the accuracy  of results changed by uploading fewer or more photos; 2) while the authors asked ChatGPT for advice on how to improve results, they did not then re-run their analysis to see if these refined prompts would have worked (in this case, you can still re-use the same photos). One of the major elements of LLM testing is in their fine tuning – it is not enough to arbitrarily choose the number of photos to upload or to only rely on a single prompt, one must play with, in this case, a range of photos and a few refined prompts to see if that would improve accuracy. As is, the paper presents a worthwhile approach, but it only represents their first try; the methods are not complete, and so it is missing some critical components in the testing of the LLM. Below are some general remarks as well as notes on these two important aspects (appropriate # of photos to upload, and refinement of the prompts). However, I think this is a novel contribution, and will probably be one of many papers like this in the near future, so I encourage the authors to incorporate the more rigorous testing so as to set a good precedent for future work (and likely more citations).

Abstract (2 instances) – “Complements” should be spelled with an “e” not an “i”: “It compliments exposure assessments…” and “…application to compliment exposure assessments…”

Introduction – not sure what “The method” is referring to: “The method is thought to enhance retrospective reporting by reflecting 57 the structure of autobiographical reporting.”

Search entire document for use of the word “complimented.” It should be “complemented.” “Complimented” with an “i” means to praise something politely. “Complemented” with an “e” means to “add to something in a way that enhances or improves it.” (e.g., “and can be complimented by observation-based inventories…” “be used to compliment and/or cross-validate” “promise for data gathering to compliment exposure assessments…”). Confirm any and all instances.

“This included…” instead of “The included…”: “The included visits to farms that prac-82 ticed conventional and organic farming methods.”

Why 8 photos per dialogue window? Did you try entering only 4 photos for example, to see if that would increase accuracy? It seems like a few options should have been tried, including just one photo per prompt to see if that improved accuracy. Chat prompts have a given token limit, so, for example, if you uploaded all 31 images, it’s likely that would have returned less accurate results. How do we know that 8 was the magic number. Why weren’t a range of photos tried (like, 1, 2, 4, 8, 15)? That would have helped elucidate any points of diminishing returns. 8 actually sounds like a lot to me; I would have tried fewer photos to at least compare results.

I do not understand why this method was not repeated with the prompt suggestions provided by ChatGPT, to see if they would indeed improve accuracy. Based on the following paragraph, the authors should have repeated the process and seen if that helped to improve accuracy: “Upon review of these findings, ChatGPT recommended several enhancements to improve 105 the reliability and applicability of this LLM-assisted method for agrichemical identifica-106 tion. First, to address “hallucination”—a common issue where LLMs may misidentify or 107 invent chemicals—future iterations of the method could employ prompts that specifically 108 instruct the LLM to “identify only chemicals visibly labeled in the image, without assump-109 tions based on partial or blurred text.” Additionally, refinement of categorization accu-110 racy could be achieved by using prompts that ask the LLM to “first identify the type (fer-111 tilizer, herbicide, insecticide, fungicide) based on specific visual and textual cues before 112 proceeding to detailed categorization.” Finally, integrating prompts that invite the LLM 113 to cross-reference detected agrichemicals with common regional products, when possible, 114 may enhance relevance and reduce the likelihood of misclassification. These adjustments 115 provide a basis for further developing this method, allowing future researchers to lever-116 age LLMs with improved precision in similarly restricted fieldwork contexts.”

Is it possible to provide a few example pictures that were analyzed by the LLM? It would be nice to see what was fed into the model for analysis.

Author Response

We thank the reviewer for these insightful observations. As a brief clarification, this manuscript was submitted as a Technical Note, with the aim of documenting the feasibility of an emerging method trialed during fieldwork in 2024. The approach reflects the state of large language model (LLM) technology at that time (ChatGPT 4o), which constrained what could be attempted in real-time field conditions. We fully acknowledge that current and future models (e.g., GPT-5) may perform differently and more effectively than the version available during data collection. Within these constraints, however, we believe that the revisions we have made provide the necessary context to address the reviewer’s concerns and demonstrate the contribution of this Technical Note as a methodological contribution. We have also added a limitations section to the revised discussion and have also provided additional detail in the discussion section regarding these points.

We have corrected all misspellings of “compliments” to “complements,” as suggested by the reviewer. We thank the reviewer for catching that spelling error. We have also changed “the included” to “this included,” as suggested.

We appreciate the suggestion to include representative images. In response, we have added Figure 1 (“Representative screenshots from agrichemical storage shed video used for LLM-assisted identification”), which illustrates the type of inputs provided to ChatGPT. For privacy reasons, we are unable to release the full image set so as to not reveal the specific producer.

We also appreciate the reviewer’s thoughtful suggestion to test a range of photo batch sizes. To clarify, we initially attempted to analyze the entire video directly but given the state of LLM technology at the time (May 2024) and the needs of the team, ChatGPT could not process the full video files as requested. As a result, we extracted screenshots and provided them iteratively while in the field. The number of screenshots uploaded to each chat window was therefore not arbitrary, but reflected the maximum number (eight) that ChatGPT could handle at once during this iterative process to provide feedback to the team in a timely manner.

While we recognize that systematically testing accuracy across varied batch sizes (e.g., 1, 2, 4, 8, 15, 31) would provide valuable insights, such an evaluation falls beyond the scope of this Technical Note. We have added a clarifying statement to the Discussion to acknowledge this limitation and to highlight future opportunities for testing batch-size effects.

Reviewer 2 Report

Comments and Suggestions for Authors

This technical note presents an innovative method that integrates ethnographic fieldwork with ChatGPT to identify agrichemicals from video footage of storage sheds in avocado farms. The approach is timely and relevant given the growing need for rapid, non-invasive methods of exposure assessment, particularly in contexts with limited researcher access and complex agrichemical use. 

Please see below for my detailed comments:

  1. The 75% accuracy rate is a useful starting point, but more rigorous validation protocols are needed. For example, provide prompts used in identification of the chemicals in Supplementary files
  2. provide more details in the Methods, such as the settings of videos, the resolution of pictures taken from the 55s video
  3. please explain possible reasons why ChatGPT failed to identify some chemicals as shown in Table 2

Author Response

We thank the reviewer for their supportive comments and constructive feedback.

We agree that systematic validation of prompt variations would strengthen this work. However, as a Technical Note our aim is to demonstrate feasibility in a concise manner rather than exhaustively test prompt structures. We identified the original prompt that was used by the investigator in the field to generate the original inventory. The verbatim prompt is now included in the Methods section of the revised manuscript.

We appreciate this suggestion and have clarified in the Methods section that the 55-second video was recorded on an iPhone 12 at 1170 × 2532 pixel resolution under indoor mixed artificial and natural lighting conditions, and screenshots of the video were generated at two-second intervals.

We appreciate the reviewer’s request for clarification on why some chemicals were not correctly identified. We have added an explanation in the new limitations paragraph noting that misclassifications or omissions most likely occurred due to blurred or partially obscured text on containers, limited recognition of certain region-specific agrichemical brands, and the use of an earlier model version of ChatGPT that may be less adept than newer iterations.

We believe these clarifications improve transparency while remaining consistent with the concise scope of a Technical Note.

Reviewer 3 Report

Comments and Suggestions for Authors

This work presents a methodological approach to agrichemical inventory documentation, applying large language model (LLM) capabilities for categorizing agrichemicals from brief video footage, using ChatGPT. 

Table 1 shows the Agrichemicals identified by ChatGPT, and Table 2 shows the Human verification of chemicals identified by ChatGPT.

Although this may seem to be an interesting study, it lacks scientific evidence and may not be suitable for publication. 

The limitation of this study is not discussed in the discussion or conclusion section. The number of references are limited with only 23 entries. 

Also, the video footage for this study is not shown in the appendix or supplementary materials.

Author Response

We thank the reviewer for this feedback. We respectfully note that this submission was prepared as a Technical Note, whose purpose is to describe and demonstrate a concise methodological approach rather than provide comprehensive empirical evidence. Within this format, our intention is to illustrate the feasibility of LLM-assisted agrichemical identification in time-restricted field settings.

We have added a statement to the discussion regarding the limitation of this technical note in providing full-scale model validation.

We have also added a sentence to the conclusion re-emphasizing that this Technical Note is a methodological demonstration.

Due to the sensitivity of storage shed contents and farmer privacy, the raw video cannot be shared in public supplementary materials. However, we have added two photographs to the revised manuscript to provide more visual context for readers.

We hope this clarifies that, within the scope of a Technical Note, the methodological contribution stands as a preliminary but useful step for subsequent, more comprehensive studies.

Round 2

Reviewer 1 Report

Comments and Suggestions for Authors

The authors have adequately addressed the strengths and limitations of the study and have provided appropriate material to cover the review comments. The additional materials outline key considerations and thoughts for improvements and future work/applications.

Reviewer 3 Report

Comments and Suggestions for Authors

Although the authors attempted to improve their work, the revision provided by the authors is not substantial enough to make a difference to this work. Therefore, my opinion on this work remained the same.